# Corneal Hysteresis, Intraocular Pressure, and Progression of Glaucoma: Time for a “Hyst-Oric” Change in Clinical Practice?

**DOI:** 10.3390/jcm11102895

**Published:** 2022-05-20

**Authors:** Patrick Murtagh, Colm O’Brien

**Affiliations:** Department of Ophthalmology, Mater Misericordiae University Hospital, Eccles Street, D07 R2WY Dublin, Ireland; cobrien@mater.ie

**Keywords:** corneal hysteresis, corneal thickness, glaucoma, progression, risk stratification

## Abstract

It is known that as people age their tissues become less compliant and the ocular structures are no different. Corneal Hysteresis (CH) is a surrogate marker for ocular compliance. Low hysteresis values are associated with optic nerve damage and visual field loss, the structural and functional components of glaucomatous optic neuropathy. Presently, a range of parameters are measured to monitor and stratify glaucoma, including intraocular pressure (IOP), central corneal thickness (CCT), optical coherence tomography (OCT) scans of the retinal nerve fibre layer (RNFL) and the ganglion cell layer (GCL), and subjective measurement such as visual fields. The purpose of this review is to summarise the current evidence that CH values area risk factor for the development of glaucoma and are a marker for its progression. The authors will explain what precisely CH is, how it can be measured, and the influence that medication and surgery can have on its value. CH is likely to play an integral role in glaucoma care and could potentially be incorporated synergistically with IOP, CCT, and visual field testing to establish risk stratification modelling and progression algorithms in glaucoma management in the future.

## 1. Introduction

The field of glaucoma is an ever-expanding one. Its intricacies and subtleties have been slowly evolving. The relationship between glaucoma and intraocular pressure (IOP) was the first association to be made but, subsequently, new and significant correlations have been discovered. The invention of the ophthalmoscope by von Helmholtz in the 1850s [1] made it possible for clinicians to document and classify optic disc appearance and correlate it to glaucoma. The advent of accurate tonometers [2] and reproducible perimetric testing [3], allowed ophthalmologists to assess the risk factors for disks and monitor disease progression. Interventions were then introduced to lower and stabilise IOP, with the introduction of surgical procedures such as the iridectomy performed by von Grafe in 1856 [4] and the use of pilocarpine in the late 1870s [5].

Glaucoma care has advanced alongside modern medicine over the past 170 years. Innovations in medications [6], the acceptance of minimally invasive glaucoma surgery (MIGS) [7], and advances in screening [8] and monitoring of the disease in terms of visual field testing and OCT scans of the RNFL and GCL, have made glaucoma a different disorder than it once was; one that is no longer “the silent thief of sight” but one that can be controlled, and the devastating visual impacts associated with it can be prevented.

Monitoring the progression and risk stratification are essential tools in glaucoma management. Additional objective markers to distinguish those with a higher probability of progression have eluded researchers and clinicians for some time but are an integral and essential part of glaucoma management. Corneal Hysteresis (CH) is a relatively new and exciting concept that could help identify those at risk of glaucoma and those who are likely to progress, and it is a property that can change with alterations in IOP. This review will explain what CH is, how it is measured, its relevance and correlation to different types of glaucoma, its significance to the progression of the disease, and the effects of changes in IOP by various methods on CH values. In this paper, we propose that it is time now to rewrite the textbooks about the value of measuring CH routinely in clinical practice.

## 2. Glaucoma

The term “glaucoma” is used to describe a group of optic neuropathies that are associated with progressive damage to retinal ganglion cells, and it is the second leading cause of permanent blindness in developed countries [9]. The insidious and irreversible nature of the disease makes early detection and screening paramount to avoid or halt the devastating functional, social, and economic impacts associated with it [10]. Glaucoma is more common in older age [11]. As the world’s older populations grow at an unprecedented rate, so too will the burden of ocular disease, with the number of people diagnosed with glaucoma expected to double from 76 million in 2020 to almost 120 million in 2040 [12].

## 3. Glaucoma and IOP

Elevated IOP is a significant risk factor for the development and progression of glaucoma [13]. IOP is currently the only modifiable parameter that clinicians adapt to control or stop glaucomatous damage [14]. The correlation between raised IOP and glaucoma is not a new one. The Ancient Greeks recognised that acute blindness could be associated with an unreactive pupil and increased tension in the eye [15]. However, it was not until the mid-19th century when von Graefe developed the first tonometer to accurately measure intraocular pressure (IOP) [16], and the hypothesis arose that lowering IOP could prevent or slow down the rate of glaucomatous progression. The advent of the Goldmann applanation tonometer [17] (based on the Imbert–Fick principles) in the 1950s allowed accurate IOP measurements to come into mainstream use and permitted the clinician to make treatment decisions based on IOP values.

## 4. Glaucoma and Central Corneal Thickness

Objective measurements that influence glaucomatous optic neuropathy are extremely important surrogate measurements of disease. Visual field assessment is a subjective measurement of disease severity and is highly dependent on the user and can sometimes be difficult for the patient to perform, which can lead to an adverse number of false positives and negatives and a lack of reproducibility [18]. Published in 2002, The Ocular Hypertension Treatment Study [19] (OHTS) revealed the baseline factors that predicted the development of primary open-angle glaucoma (POAG) from ocular hypertension (OHT), the most significant being central corneal thickness (CCT). It was postulated that having a thicker cornea was protective against glaucomatous optic neuropathy. This finding lead to a risk stratification algorithm for patients with high IOP and no signs of optic nerve damage. These results were echoed in the European Glaucoma Prevention Study (EGPS) [20]. Subsequent work from the ocular hypertension treatment study group elucidated that for every 40 μm of corneal thinning, a twofold increase in the risk of developing GON was seen over a five-year period [21]. The underlying aetiology of this finding is still uncertain, and it is ambiguous whether it is secondary to its influence on IOP measurement, whether it is related to an intrinsic corneal characteristic and hence ocular tissue property, or a combination of both [22].

## 5. Age and Ocular Stiffness

Advancing age is a major risk factor for glaucoma progression [23]. Historically, the presumed mechanism of action was IOP, which resulted in mechanical stress on the optic nerve leading to ganglion cell death. Characteristic visual field loss and a cupped optic nerve head are hallmark signs of glaucomatous optic neuropathy (GON). The increase in IOP was mainly believed to be secondary to the outflow obstruction [24]. This mechanism has since been brought into question. There is a growing body of evidence that supports a decrease in the compliance of ocular structures and an increase in stiffness [25]. The structures affected include the trabecular meshwork (TM) and Schlemm’s canal [26], the peripapillary sclera, and the lamina cribosa [27]. The mechanism underlying this stiffening includes extracellular matrix remodelling and fibrosis, initiated by the cytokine transforming growth factor-beta and oxidative stress [28]. The decrease in compliance of the ocular tissues leads to (1) outflow resistance in the TM due to a decrease in height, an increase in thickness [29], and (2) increased optic nerve stress due to increased rigidity in the lamina cribosa and peripapillary sclera [30]. A method to accurately measure the rigidity of these structures has been a challenge; however, a useful surrogate marker has come to the fore in recent times.

## 6. Hysteresis and the Cornea

Hysteresis is scientifically defined as a lag between the input and output in a system upon a change of direction [31]. It is dependent on the state of the system and its history [32]. It can reflect the intrinsic property of a material, and in biological materials, it can indicate its biomechanical qualities. The cornea can be characterised by its inherent behaviour. It has viscoelastic properties [33] and this can be reflected in the measurement of an applied force on the cornea, and its subsequent action and reaction to the said force [34]. As ophthalmologists, we are very familiar with viscoelastic materials as we routinely utilise them intraoperatively to maintain space and protect intraocular structures [35]. As the name suggests, a viscoelastic material is one that displays both viscous and elastic traits and is able to incorporate mechanical stress and disperse it sufficiently [36]. This dispersion is akin to a biological shock absorber and does not transmit force or allow it to accumulate in one specific area. It is, therefore, believed that the greater the value of this shock absorption or hysteresis, the greater the intrinsic ability of the ocular structures to deal with applied force or stress and, therefore, the lower the likelihood of nerve damage due to increased strain on the optic nerve head in the area of the lamina cribosa. Numerous studies have demonstrated that eyes with lower hysteresis values had faster rates of visual field loss than those with higher hysteresis values [37,38,39]. Corneal hysteresis may be a surrogate marker for hysteresis values elsewhere, specifically in the peripapillary sclera and the trabecular meshwork.

## 7. Corneal Hysteresis Measurement

Currently, there are two devices on the market used to measure corneal hysteresis. The first is known as the ocular response analyser (ORA; Reichert Inc., Depew, NY, USA). The ORA uses a quick jet of air to indent the cornea and an electro-optical system is used to measure the applanation pressure, once when the cornea is displaced inward, and again when it is displaced outward. The cornea has viscoelastic properties (as mentioned above) and therefore it resists inward movement caused by the air pulse and reverts to its primary position due to its elastic nature. There is a delay between these applanation events. The first inward applanation pressure is termed P1 and the second, or outward pressure event, is classified as P2. The Goldmann-correlated IOP (IOPg) is the average of these two values. The difference between P1 and P2 is known as the corneal hysteresis (CH) value [40]. Two other parameters, namely the corneal-compensated IOP (IOPcc) and a corneal resistance factor (CRF), can also be derived from the ORA data. IOPcc is a pressure measurement that utilises CH to give a pressure value that is considered to be less influenced by intrinsic corneal properties, e.g., central corneal thickness [41]. CRF is a depiction of overall corneal resistance and is algorithmically calculated [42]. Clinical use for these accessory criteria is yet to be elucidated [43].

The second device is the Corneal Visualization Scheimpflug Technology tonometer (Corvis ST; Oculus, Wetzlar, Germany). Similar to the ORA, the Corvis ST utilises an air pulse, but a high-speed Scheimpflug camera is used to calculate corneal movement. It records the cornea’s reaction to the air pulse and the camera can take up to 4300 images per second. A video of 140 images taken 31 ms after the onset of the air pulse is used to provide a detailed analysis of the biomechanical properties of the cornea. A biomechanically corrected IOP (bIOP) is calculated, analogous to the IOPcc above [44]. The Corvis ST technically does not provide corneal hysteresis results, but bIOP is influenced by the effects of CH and is, therefore, a useful substitute marker. The parameters obtained by the Corvis ST are currently not directly comparable with those obtained by the ORA [45].

CH values are measured in millimetres of mercury (mmHg). The values are repeatable and vary among patients, and studies have found that CH values in non-pathological eyes have an average value of between 10.2 and 10.7 mmHg [46]. CH values display ethnic variations, with a study by Haseltine et al. [47] demonstrating that in non-glaucomatous eyes, Black, Hispanic, and Caucasian subjects have average CH values of 8.7 mmHg, 9.4 mmHg, and 9.8 mmHg, respectively. CH values are independent of other corneal measurements such as radius, refractive error, or IOP [48]. CH and CCT are positively associated [49].

## 8. Corneal Hysteresis and Glaucoma

Glaucoma can have devastating effects on a patient from a social, economic, and health point of view. There is a significant spectrum of diseases under the umbrella term that is glaucoma, including primary open-angle glaucoma (POAG), closed-angle glaucoma, pigment-dispersion glaucoma, pseudoexfoliative glaucoma, and normal-tension glaucoma (NTG), to name but a few. The Holy Grail of glaucoma management is to determine which patients have a greater likelihood of progressing from ocular hypertension to glaucomatous optic neuropathy and, in patients who already have glaucoma, which are more likely to deteriorate quickly. With the advent of more widespread use of CH measurement, this may soon become a reality. (Figure 1).

## 9. Primary Open-Angle Glaucoma (POAG)

In simple terms, it is known that corneal hysteresis is lower in eyes with POAG than in eyes without glaucoma [50]. In a prospective cross-sectional study in 2010, Anand et al. [51] examined 117 POAG patients with asymmetric visual fields. This asymmetric POAG was associated with the corresponding asymmetry in ORA parameters but not in CCT or IOP. Lower CH was an independent risk factor for the eye with a worse visual field, irrespective of its pressure. A study by Dana et al. in 2015 [52] demonstrated a statistically significant, positive correlation between Visual Field Index (VFI) and CH, with lower CH values correlating with a lower VFI on Humphrey Visual Field testing. In a recent study published in 2021 by Jiménez-Santos et al. [53], 1573 patients from a previous cohort study with POAG were analysed in terms of glaucoma progression with respect to multiple baseline parameters including CH and CCT. It was observed that patients without progression had higher CH values and higher CCT. Using multivariate analysis, it was revealed that for every 1 mmHg reduction in CH measurement, an increase of 2.13 in terms of the hazard ratio for the risk of progression was conferred. The authors concluded that CH was considered to be a risk factor in early POAG and that CCT and CH at higher values work synergistically to slow the rate of progression. However, not all studies demonstrated this combined effect between the parameters, with Sullivan-Mee et al. [54] showing that after multivariate analysis, CH was the only factor that continued to discriminate between normal and glaucomatous eyes.

## 10. Angle-Closure Glaucoma (ACG)

Sun et al. [55] revealed that CH values were significantly lower in patients with chronic primary ACG as opposed to age-matched non-pathologic controls, with the presenting CH value in the glaucoma group measuring 6.83 ± 2.08 mmHg as opposed to the control eyes, which had an average CH value of 10.59 ± 1.38 mmHg. Another prospective observational study was undertaken by Narayanaswamy [56] et al., who examined 131 patients with primary ACG from a cohort of 443 Chinese patients. When confounding factors were adjusted, CH values were significantly lower in primary ACG eyes compared to normal eyes (9.4 mmHg versus 10.1 mmHg). However, other studies have found this relationship to be inconsistent, with a study by Nongpiur et al. revealing a lack of correlation between CH values and severity of disease in chronic angle-closure glaucoma [57] patients.

## 11. Pseudoexfoliation Glaucoma (PXFG)

A retrospective study undertaken in 2011 in Sweden [58] examined 90 patients for CH values; 30 with POAG, 30 with pseudoexfoliation glaucoma (PFXG), and 30 without glaucoma. The patients were also age matched. The results indicated that CH values were significantly lower in PFXG patients in comparison to both the POAG eyes and the non-glaucomatous eyes. Mean CH values in normal, POAG, and PXFG eyes were 9.8 ± 1.6 mmHg, 9.0 ± 1.9 mmHg, and 8.0 ± 1.5 mmHg, respectively. Yazgan et al. in 2014 [59] compared patients with pseudoexfoliation syndrome (PFXS), PFXG, and controls and revealed that CH was decreased in both PFXS and PFXG patients but to a greater degree in the PFXG patients. Yenerel et al. [60] showed that both CH and CRF values were lower in patients with both unilateral and bilateral PFXG. Interestingly, another study examining the use of the Corvis ST in measuring corneal biomarkers [61] showed no difference between eyes with PXG, POAG, and healthy controls after adjusting for IOP.

## 12. Normal-Tension Glaucoma (NTG)

Normal-tension glaucoma (NTG) is defined as evidence of glaucomatous optic neuropathy in eyes with an IOP of 21 mmHg or less. Studies have estimated that from a global perspective, as many as 30–50% of glaucoma patients may have IOP considered to be within the normal range [62]. Park et al. [63] analysed 95 NTG patients and evaluated them with respect to 93 patients without glaucoma. They concluded that patients with NTG had lower CH values than those in the normal group. They categorised their patients as normal, early NTG, and advanced NTG and the CH values were 10.83 ± 1.60, 10.56 ± 1.44, and 9.78 ± 1.52, respectively, with more advanced disease correlating with a lower CH value. They also determined that CH value alone remained statistically significantly associated with optic nerve head parameters (such as rim area and volume and cup-disc ratio) after adjusting for other confounding factors. They surmised that CH has a greater influence on structural biomarkers than CCT in NTG patients. The findings that NTG eyes have lower CH values than normal eyes have been echoed in numerous other studies [64,65,66].

Morita et al. [67] noted that IOPcc is significantly higher in eyes with NTG than in normal eyes, and Ehrlich et al. [68] noted that in comparison to eyes with POAG, there was a greater discrepancy seen between IOP measured with Goldmann Applanation Tonometry (GAT) and ccIOP with NTG eyes. A study by Hong et al. in 2016 [69] looked at the rate of progression in NTG patients and stated that eyes with a lower CH value and a higher ccIOP were likely to progress quicker than those with either higher CH values or lower ccIOP values. There was a substantial difference noted between GAT and ccIOP in these patients. They concluded that it is likely that GAT underestimates IOP in these patients and that ccIOP is a more accurate representation of actual IOP.

CCT is routinely used in NTG as a risk stratification tool. Previous studies have found that in patients with NTG, their CCT is thinner in comparison to both POAG eyes and normal eyes [70]. However, more recent studies [71] have found that CH and CRF are more robust predictors of progression in NTG than CCT.

Please see Table 1 for a summary of CH values in terms of glaucoma detection by subtype.

## 13. Corneal Hysteresis and Glaucomatous Progression

The first association between CH and visual field progression was made by Congdon et al. [72] in 2006. This was an observational study which included 230 patients with either POAG (85%) or suspected glaucoma (15%). The cohort underwent routine baseline evaluations and, subsequent to multivariate generalised estimating equation modelling, a lower CH was associated with greater visual field progression, an association that was not apparent for CCT. Susanna et al. [38] performed a prospective observational study on 199 patients who were suspected of having glaucoma and were followed for an average of 3.9 years. Glaucoma progression was defined as Glaucoma Hemifield Test outside normal limits or a Pattern Standard Deviation (PSD) of <5% on three consecutive automated perimetry tests. Of the 54 eyes that developed repeatable visual field defects on follow up, their CH values were significantly lower than those whose fields remained static. They concluded that lower CH values were associated with a higher risk of developing glaucomatous visual field defects over time. De Moraes et al. [73] examined the relationship between CH, CCF, and visual field progression in terms of decibels lost per year. They deduced that eyes that have greater progression of their visual field had lower CH and CCT values and that eyes that had the greatest number of decibels lost had lower CH values. Medeiros et al. [37] examined CH values and the loss of the Visual Field Index (VFI) over time. Out of the 68 patients with known glaucoma, 114 eyes were followed for an average of 4 years. The results revealed that CH had a significant effect on progression, more so than IOP and CCT, and that for each 1 mmhg lower CH value conferred an associated risk of 0.25%/year faster rate of visual field progression over time (*p* < 0.001). In a more recent study by Estrela et al. [74], the asymmetries between glaucoma progression and the asymmetries in corneal properties were examined in a prospective study of 126 binocular glaucoma patients. Visual field progression was determined by a change in mean deviation (MD) on standard automated perimetry testing (SAP). The only corneal property of those measured (including CCT and IOP) that had a statistically significant association with an asymmetry of SAP MD rates was the difference in CH values. This remained the situation even after multivariate analysis to void confounding factors including age, race, CCT, and IOP. The authors predicted that for each 1 mmHg change in CH value in eyes of the same subject, a 34% increase in the variance of MD rates could be observed.

Subjective visual field progression was not the only parameter analysed in the studies. Zhang et al. [75] examined the relationship between CH and retinal nerve fibre layer (RNFL) thinning in a prospective follow up of 186 eyes of 133 patients. They were followed for an average of 3.8 years with a median follow up of 9 visits. Measurements of RNFL were obtained using spectral-domain optical coherence tomography and potential confounding factors were adjusted for. The authors determined that CH had a significant effect on RNFL thinning, with each 1 mmHg lower CH value being associated with a 0.13 μm/year faster rate of RNFL decline (*p* = 0.011).

A recent study by Kamalipour [76] et al. examined, in a prospective longitudinal study, CH as a risk factor for the progression of the central visual field in a cohort of glaucoma patients. Out of 143 patients, 248 eyes were examined using HVF 24-2 and 10-2 over an average of 4.8 years. Logistic regression analysis was utilised to determine the characteristics that would influence progression on a 10-2 field. The authors showed that lower CH values were associated with a statistically significant, albeit small, increased risk of central visual field progression. However, the central visual field has a huge impact on a patient’s quality of life and so the paper surmised that CH should be considered by clinicians as a risk stratification parameter in glaucoma.

Table 2 summarises the main findings of the above studies with respect to CH values and glaucoma progression.

## 14. Effect of IOP Reduction on Hysteresis

CH is a dynamic property of the cornea. As previously stated, lower CH values are associated with both an increased likelihood of developing glaucoma and a faster rate of progression. Studies have shown that by lowering the intraocular pressure, CH values can increase. In a retrospective review by Agarwal et al. in 2012 [77], 57 patients with POAG were analysed by ORA at baseline and follow up, subsequent to the commencement of a prostaglandin analogue for the treatment of their disease. It was seen that IOP was reduced by an average of 3.2 mmHg, which corresponded to an increase in the CH value of 0.5 mmHg. It was demonstrated that baseline CH, and not baseline CCT, was a significant predictor of IOP reduction, with a lower baseline CH associated with a greater reduction in IOP. The effect of topical prostaglandin analogues on CH was also demonstrated by Tsikripis et al. [78] in their study examining the influence that lower IOP had on CCT biomechanical markers including CH, CRF, and CCT. Out of the 108 eyes that were included in this study, 66 were treated with latanoprost solely and the remaining 42 eyes were treated with a combination of latanoprost and timolol. It was seen that by using topical prostaglandin analogues, the IOP values decreased with a corresponding increase in the CH and CCT values, with a range for the increase in CH values of 0.4–0.7 mmHg and 0.65–0.95 mmHg for the latanoprost and the latanoprost/timolol group, respectively.

Topical medication is not the only treatment modality for which a reduction in IOP influences CH values. A study by Pillunat [79] et al. in 2016 examined 52 eyes of 52 patients with medically uncontrolled glaucoma and performed Selective Laser Trabeculoplasty (SLT) to control IOP. They found that Goldmann-correlated IOP decreased statistically significantly from 18.0 ± 6.4 to 14.8 ± 3.8 mmHg and IOPcc from 20.2 ± 6.5 to 16.7 ± 3.4 mmHg (*p* < 0.001). CH increased from 8.53 ± 2.03 to 9.12 ± 1.83 mmHg (*p* = 0.028) and CRF decreased from 9.58 ± 2.18 to 9.1 ± 2.1 mmHg (*p* = 0.037), which was statistically significant. However, in covariance analysis, by correcting CH and CRF for the impact of IOP reduction, the CH and CRF values remained unchanged. The authors concluded that SLT may not change the corneal biomechanical properties, as these changes may solely be explained by changes in IOP. However, in 2013, Hirneiß et al. [80] analysed 68 patients with open-angle glaucoma that were insufficiently controlled by topical medications and hence underwent SLT for IOP control. They were examining the predictive values of corneal biomarkers for IOP reduction post SLT. A total of 68 patients with open-angle glaucoma (OAG) were followed for 12 months after the procedure. Linear regression analysis revealed that both CH and CRF alongside baseline IOP correlated significantly with IOP reduction. It was surmised by the authors that the original IOP, CH, and CRF values were significant predictors of the IOP-lowering effect of SLT in medically resistant OAG.

Surgery to reduce IOP has also shown to positively impact CH values. A study by Pakravan et al. [81] examined the 89 eyes of 89 patients with ORA before and three months after either trabeculectomy and mitomycin C (MMC) (23 eyes), phacotrabeculectomy + MMC (23 eyes), Ahmed valve implantation (17 eyes), or phacoemulsification alone (26 non-glaucomatous eyes). Their findings revealed that CH was lower in glaucomatous vs. non-glaucomatous eyes. Three months post-surgery, it was shown that CH values increased in the trabeculectomy and MMC group, the phacotrabeculectomy, and the Ahmed value group by 2.16, 2.29, and 2.30 mmHg, respectively. However, an increase in CH of only 0.11 mmHg was seen in the post-phacoemulsification only eyes. The increase in CH values was most significant in the eyes where IOP was decreased by 10 mmHg or more. Fujino et al. [82] examined 24 eyes of 19 patients with POAG who underwent trabeculectomy and recorded CH values before and after surgery in conjunction with Humphrey visual field testing to assess progression. Their modelling for progression based on mean deviation on-field testing demonstrated that only CH values had a positive correlation coefficient for the rate of change. In 2019, Sorkhabi et al. [83] examined 32 eyes of 32 patients, 17 of whom had PXFG and the remaining 15 had POAG. All patients underwent trabeculectomy and MMC for uncontrolled glaucoma and ORA parameters were recorded at baseline and 3 months post procedure. The authors found that the mean CH values were lower in the PXFG group than in the POAG group at baseline. The CH values markedly increased in the PXFG group and modestly increased in the POAG group post-surgery (5.66 ± 1.13 to 6.69 ± 0.78 and 7.49 ± 0.88 to 8.23 ± 1.09). The authors also noted that there was a significant relationship between CH and IOPg changes in both the PXFG and POAG groups.

Contrary to the above studies, a recent paper by Pillunat et al. [84] demonstrated that when confounding factors were adjusted, the corneal biomechanical properties were not altered post trabeculectomy. In this study, 35 eyes of 35 patients undergoing trabeculectomy were enrolled and it was noted that the changes in CH values before and after trabeculectomy were not statistically significant.

## 15. Limitations

The acquisition of CH measurements in clinical practice is currently not mainstream. Not all units have access to an ORA and presently ophthalmologists are largely unfamiliar with the values and how to interpret them.

The precise mechanisms by which CH values affect glaucoma detection and progression are unclear apart from them being a surrogate marker for ocular stiffness. Is there a vascular or an ischaemic factor underlying these mechanisms? A recent review by Hopkins et al. [27] proposed a three-stage tissue stiffness model incorporating integrin-mediated mechanotransduction that leads to extracellular matrix remodelling and fibrosis and, in turn, to diminishing contractile ability of the lamina cribosa. Further research should be undertaken to address these obvious gaps in knowledge and elucidate the intrinsic process by which it works.

## 16. Conclusions

The contribution that CH will eventually have to glaucoma care has yet to be fully appreciated. This review has summarised its effects and relevance on different types of glaucoma, how its values can fluctuate with alterations in treatment, and its significance in monitoring progression. It has become apparent that corneal behaviour is a more important parameter than its thickness, but the authors believe that a combination of IOP, CCT, and CH can be utilised to create a risk stratification model for glaucoma. Undoubtedly, additional investigation is needed in this field and with it, the importance of CH in diagnosing and monitoring this potentially devastating disease will likely come to the fore.

## Figures and Tables

**Figure 1 jcm-11-02895-f001:**
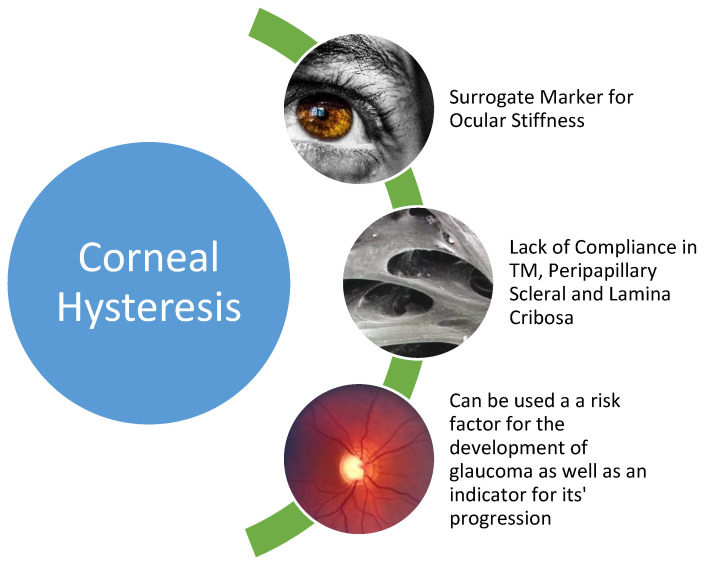
Corneal hysteresis is a surrogate marker for glaucoma and can be utilised to risk-stratify both those who are in danger of developing the disease and also those who are likely to progress (TM: trabecular meshwork).

**Table 1 jcm-11-02895-t001:** Summary of main findings of CH studies in terms of glaucoma detection divided by subtype of glaucoma.

Study Lead Author and Year	Study Type	Number of Patients	Main Finding
** *POAG* **			
Sullivan-Mee et al., 2008 [54]	Retrospective	298	CH Values are useful in differentiating between patients with and without POAG.
Anand et al., 2010 [51]	Prospective	117	Asymmetric POAG was associated with asymmetry in ORA parameters. Lower CH was associated with more advanced glaucomatous disease.
Dana et al., 2015 [52]	Observational	55	Positive, statistically significant correlation between CH values and VFI. Lower CH Values are associated with lower VFI.
Jiménez-Santos et al., 2021 [53]	Cohort	1573	CH can be considered as a risk factor of progression in early-stage POAG.
** *ACG* **			
Sun et al., 2009 [55]	Prospective	80	CH was significantly lower in chronic PACG patients.
Narayanaswamy et al., 2011 [56]	Prospective	443	Corneal hysteresis was lower in eyes with glaucoma and after adjusting for confounding factors, lower CH values was found in PACG eyes.
Nongpiur et al., 2015 [57]	Prospective	204	Severity of glaucoma in PACG is *not* associated with lower CH values.
** *PXFG* **			
Ayala et al., 2011 [58]	Retrospective	90	CH was significantly lower in PXFG patients than in POAG normal patients, but no significance was found between the POAG and the normal group.
Yenerel et al., 2011 [60]	Prospective	52	CH reduces in patients with both unilateral and bilateral PEX.
Yazgan et al., 2015 [59]	Prospective	118	CH values were decreased in patients with PXFG, more so than in patients with solely PEX.
Pradhan et al., 2020 [61]	Prospective	66	After adjusting for IOP, CH values for normal eyes, POAG eyes and PEX eyes did not differ.
** *NTG* **			
Morita et al., 2012 [67]	Prospective	166	IOPcc and CH values were significantly higher in NTG eyes than in normal eyes.
Ehrlich et al., 2012 [68]	Retrospective	614	Compared to GAT, IOPcc may be a superior test in the evaluation of glaucoma as it may account for measurement errors induced by corneal biomechanics.
Hong et al., 2016 [69]	Prospective	56	Higher IOPcc and lower CH are associated with VF progression in NTG patients.
Park et al., 2018 [63]	Retrospective	188	Lower CH values are associated with a smaller rim area and volume, thinner RNFL, and a larger cup disc ratio after adjusting for CCT, age, IOP, and disc size.

POAG = Primary Open-Angle Glaucoma, ACG = Angle-Closure Glaucoma, PACG = primary angle-closure glaucoma, CH = Corneal Hysteresis, PXFG = Pseudoexfoliative Glaucoma, PEX = Pseudoexfoliation Syndrome, NTG = Normal-Tension Glaucoma, IOPcc = corneal-compensated intraocular pressure, IOP = intraocular pressure, ORA = Ocular Response Analyser, VFI = Visual Field Index, VF = Visual Field.

**Table 2 jcm-11-02895-t002:** Summary of main findings of CH studies in terms of glaucoma progression.

Study Lead Author and Year	Study Type	Number of Patients	Main Finding
Congdon et al., 2006 [72]	Observational	230	Lower CH values were associated with visual field progression.
De Moraes et al., 2012 [73]	Prospective	153	High correlation between VF progression and CH values.
Medeiros et al., 2013 [37]	Prospective	68	Eyes with lower CH had faster rates of visual field loss than those with higher CH.
Zhang et al., 2016 [75]	Prospective	133	Lower CH was significantly associated with faster rates of RNFL loss over time.
Susanna et al., 2018 [38]	Prospective	199	Baseline lower CH measurements were significantly associated with an increased risk of developing glaucomatous visual field defects over time.
Estrela et al., 2020 [74]	Prospective	126	In eyes with asymmetric CH values, there was an associated asymmetric VF progression, with lower CH values associated with greater rates of progression
Kamalipour et al., 2022 [76]	Prospective	143	Lower CH values were associated with a greater risk of progression on 10-2 VF

CH = Corneal Hysteresis, RNFL = Retinal Nerve Fibre Layer, VF = Visual Field.

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
