# Peer review of "Corneal Hysteresis, Intraocular Pressure, and Progression of Glaucoma: Time for a “Hyst-Oric” Change in Clinical Practice?"

_jcm, 2022, doi:10.3390/jcm11102895_

Round 1

Reviewer 1 Report

I very much enjoyed reading the review by Murtagh and Obrien, which quite thoroughly explain the various studies on the use and clinical applications of corneal hysteresis.

I only have a few suggestions and comments regarding their manuscript:

  1. the authors provide a short review of the history of glaucoma and the developments made since the disease was first recognized. However, I found it a bit bewildering that they decided to mention MIGS procedures and failed to mention the use of OCT in both diagnosis and detection of progression. especially since they later refer to the use of OCT in relation to CH.
  2. I was a little confused as to why the authors chose to describe the relationship between CH and glaucoma by looking at the different subtype of glaucoma. Should IOP measurements be interpreted differently based on whether its POAG or PXFG? all these examples can be grouped together which will only further strengthen the evidence for the use of CH in evaluating glaucoma patients.
  3. Further to the last point, the authors do look at all sub-types as one again when referring to progression (somewhat repeating the same information). In other words, I feel that the manuscript might be reorganized to better highlight the use of CH in detection of glaucoma.
  4. Adding a table that summarizes the data regarding detection and another regarding progression would make it much easier for the reader to understand the value of CH.
  5. The title and abstract of the paper are a bit misleading. In the conclusion, the authors themselves state that despite the numerous publications so far, more evidence is still needed to fully understand the use of CH in glaucoma detection and progression. I think both the abstract and title should be less definitive.

Reviewer 2 Report

JCM aims to publish "clinical and pre-clinical research" and in the case of reviews it asks for "identifying current gaps or problems and provide recommendations for future research". This paper is more of a review of knowledge that aims to recommend the use of this technique, but it lacks an interpretation of what it offers and how it offers it, as well as a projection into the future".

 The paper does not describe a hypothesis as to the mechanism by which CH influences glaucomatous damage. Is the essential point that CH distorts the interpretation of the intraocular pressure? Does CH indicate that optic nerve tissues have similar characteristics that facilitate axonal damage? Does this occur purely by mechanical effect? By vascular compression? Nowhere in the text is there any reference to the relationship between mechanical and vascular factors. The importance of ischaemia and perfusion mechanisms in the origin of axonal damage, their relation to mechanical processes or to the imbalance between ocular pressure and intracranial pressure etc., are not explained. All this should be explained, in order to define the limitations, identify the problems, and make a projection towards future research.

As minor issues I think that the introduction should be less detailed (von Helmhotz, Goldmann, Graefe etc...) or, if the authors prefer to keep it, a little more complete, because it forgets the progress of the last decades in morphological and vascular analysis.

Finally, it cannot be stated that "Glaucoma is, generally, a disease of the elderly". Congenital glaucoma also exists. And so does pigmentary glaucoma and glaucoma secondary to the use of corticosteroids, etc.

Nor can it be stated that "The most significant risk factor for glaucoma is elevated IOP".  What about normal-tension glaucoma?  At the moment only the following sentence seems defensible: "IOP is currently the only modifiable parameter that clinicians adapt to control or stop glaucomatous damage".

In short: If the editors consider that a literature review on this aspect falls within the aims of the journal, I recommend that the authors revise some of the statements they make, fill in some gaps, propose an explanation for the mechanisms by which CH is related to glaucomatous damage, and make recommendations for future research based on all of this.

Reviewer 3 Report

This is a very good review of the state of knowledge concerning hysteresis and its relation to glaucoma detection and management.

I would suggest that to make it more comprehensive you include details of the normal range found in normal individuals and how that varies by ethnicity

Round 2

Reviewer 1 Report

I cannot seem to find a response to my comments in the authors cover letter

Furthermore, I found it difficult to understand what was added or removed in the revised manuscript 

Please correct and resubmit 

Reviewer 2 Report

----
